# Night Warming Has Mixed Effects on the Development of the Fall Armyworm, *Spodoptera frugiperda* (Lepidoptera, Noctuidae), in Southern China

**DOI:** 10.3390/insects15030180

**Published:** 2024-03-07

**Authors:** Yangcheng Xu, Haipeng Chi, Mingyue Shi, Zhaozhi Lu, Myron P. Zalucki

**Affiliations:** 1School of Life Science, Institutes of Life Science and Green Development, Hebei University, Baoding 071002, China; 2Shandong Engineering Research Center for Environment-Friendly Agricultural Pest Management, College of Plant Health and Medicine, Qingdao Agricultural University, Qingdao 266109, China; 3School of the Environment, The University of Queensland, Brisbane 4072, Australia

**Keywords:** FAW, nighttime warming, accelerated development, degree days

## Abstract

**Simple Summary:**

The fall armyworm, *Spodoptera frugiperda* (Lepidoptera, Noctuidae), invaded China in 2019. The species has been established as a year-round breeding population in most of the southern provinces which are the main winter maize planting areas. The effect of rising nighttime temperatures, driven by the changing climate in this region, on the growth and development of *S. frugiperda* is unclear. Results of this study show that the survival rate of larvae and pupae significantly declined with daytime temperatures declining and the nighttime temperature range increasing. Development rate accelerated along with the increasing daytime temperatures and nighttime temperatures, except for the intermediate treatments (daytime temperature 24 °C and diurnal range from 2–6 °C). Predictions of FAW development and warnings to local farmers need to be adjusted to take into account more rapid development when nighttime temperatures increase in the warming climate.

**Abstract:**

The Fall Armyworm, *Spodoptera frugiperda* (Lepidoptera, Noctuidae), is a serious migratory pest. After invading China in 2019, the species was established as a year-round breeding population in most of the southern provinces. The area of winter maize in this region has been increasing due to the huge demand of fresh maize consumption, which is potentially at risk from this invasive pest, although the growth and development of *S. frugiperda* in the region’s changing climate is unclear, particularly with rising temperatures at night. Here, we used the highest daytime temperatures of 27 °C, 24 °C, 20 °C and decreased these by 2, 4 and 6 °C to reflect the range of nighttime temperatures indicative of winter conditions in a warming climate to evaluate the effect of increasing night temperatures on the growth and development of *S. frugiperda*. Results show that the survival of larvae and pupae significantly declined with daytime temperatures declining and the nighttime temperature range increasing. Significant developmental effects were observed across all daytime–nighttime temperature treatments, except for adults. Additionally, there were significant interaction effects for all stages, except the egg stage, and generation time. The development rate increased with the increasing daytime temperatures and nighttime temperatures, except for the intermediate treatments (Group II). The uniformity of pupation and emergence times were higher under high daytime temperatures and nighttime temperature treatments. Predictions of FAW development and warnings to local farmers need to be adjusted to take into account the more rapid development when nighttime temperatures increase in the warming climate. These results will support decision makers in developing long-term management strategies for FAW in southern China.

## 1. Introduction

From its first detection in China during January 2019, the Fall Armyworm (FAW), *Spodoptera frugiperda* (Lepidoptera, Noctuidae), a serious migratory agricultural pest, had spread to 26 provinces (autonomous regions and municipalities) by October 2019, damaging more than 106.5 million hectares of maize fields that year [1]. FAW effectively completed all the stages of the invasion process during 2019; from introduction, colonization, and incubation, to spreading and causing damage. South of 28° N in China, the species can continue to reproduce and it can survive winter between 28°–31° N [1].

FAW is a new major pest in Chinese agriculture [2] as it is polyphagous, having been recorded on more than 353 plants, of which 180 were agricultural crops [3,4,5], including maize, sorghum, and peanuts; for more, see [6]. Two genetically distinct sub-populations, or strains, are recognized: “corn-strain” and “rice-strain”, based on their host preference, mating behavior and sex pheromone [7,8]. From a genomic analysis of SNPs and gene sequences of 318 *S. frugiperda* moths collected from 13 provinces, researchers concluded that the *S. frugiperda* strain that invaded China was derived from “rice-strain” female parent and “corn-strain” male parent [9,10]. Most field populations have been recorded on maize and other Poaceae, as has been found in Africa [11], India [12] and Australia [6]. Some 83% of FAW moths trapped on Yongxing island had fed on C4 plants such as maize, and not rice [13]. The *S. frugiperda* strain in China mainly damages maize, especially fresh maize. Larvae can tunnel and destroy the maize ear during the milk maturation stage [14]. FAW has the potential to reduce annual maize production by 21–53% in the absence of pest management [11]. Guangdong, Guangxi and Hainan provinces in southern China are suitable for year-round populations of FAW [15], with about 300,000 hectares of maize [16] at risk.

Temperature influences the life of an ectotherm, determining its body temperature and, consequently, influencing the rate of physiological functions affecting growth, development, and fitness components [17,18,19]. The average global air temperature at the surface has risen substantially during the 20th century [20], with nighttime temperature increases greater than day temperatures [21,22]. The low temperature phase of the thermocycle (daily cycle of temperature) appears to play a major role in determining an insect’s performance [23]. Here, our primary goal was to evaluate the effect of increasing nighttime temperatures on the growth and development of *S. frugiperda* relevant to the maize planting area in southern China. Most work on the effects of temperature on insects uses constant temperatures, and FAW is no exception [24,25,26,27]. Here, we test the effects of daytime–nighttime temperature regimes relevant to the maize growing area of southern China on *S. frugiperda* by evaluating various life history traits. Constant temperature studies do not predict their development in our changing temperatures. Under some fluctuations, FAW develops faster than expected. Our results will better inform attempts at predicting the timing of FAW stages in the field.

## 2. Materials and Methods

### 2.1. Insect Rearing

The *S. frugiperda* used in this study was acquired from the Tobacco Research Institute of Beijing Academy of Agricultural Sciences, reared at 27 ± 1 °C, humidity 70 ± 5% for more than 20 generations in the insectary. The first three (1–3) larval stages were fed fresh maize seedlings (variety: ZD958, height: 30 cm) in cages (40 cm × 30 cm × 45 cm), and the late larval stages (4–6) were fed an artificial diet (Appendix A) individually in 6 well-plates [28]. Pupae were kept in Petri dishes until emergence. Adults were kept in mesh cages (40 cm × 30 cm × 45 cm) for mating and oviposition, and a honey–water solution (10%) was provided.

Egg masses (n = 45), each with at least 30 eggs (≤12 h old), were placed on fresh maize leaves in 9 climate chambers (5 egg masses/chamber). Larvae were transferred (twice a day) to fresh maize leaves when the eggs hatched, and the time of each transfer was noted. Third instar larvae were reared individually on fresh maize leaves in finger tubes (2 cm diameter) covered with nylon gauze (120 mesh) until they pupated. Pupae were moved to Petri dishes (30%) filled with soil until emergence. We checked the insects twice a day and recorded the instar. Emerging adults were placed in cages (40 cm × 30 cm × 45 cm), with honey water (10%), and the number of living insects was recorded daily until all the moths had died.

### 2.2. Experimental Design

FAW can survive and develop in the ambient temperature range of 14–32 °C, although mortality is high at temperatures below 18 °C [29]. Temperatures above 32 °C are less suitable for the species, and the largest pupal body weight and highest fecundity were recorded at 20–28 °C, suggesting these conditions are optimal [26,29,30,31]. Daytime temperature data over five years (2018–2022) were downloaded from a weather website https://tianqi.2345.com/ (accessed on 10 June 2023). Daytime maximum temperature and diurnal temperature variation/ range during the maize growing season, October to March, was summarized for southern provinces in China (Figure 1A). Based on the results, we selected three high daytime temperatures of 27 °C, 24 °C, 20 °C, and three nighttime temperature regimes that differed by 6 °C to 4 °C and 2 °C to give three treatment groups; Group I: 27–21 °C, 27–23 °C, 27–25 °C; Group II: 24–18 °C, 24–20 °C, 24–22 °C; Group III: 20–14 °C, 20–16 °C, 20–18 °C (Figure 1B). The temperature cycle was set to 12 h day temperature and 12 h night. The time needed for the temperature to change from one to the other was 30 min. The climate chambers’ temperatures were confirmed with thermometers and checked daily during the study.

All the studies were conducted simultaneously in 9 climate chambers (Model: RGC-500B, Anhui Youke Instrument and Equipment Company): the variation in temperature was ±1 °C and ±3–7% in humidity. We set the climate chambers on the temperature change mode with the following photoperiod: L:D = 14 h/10 h, humidity: 75%, light strength:15,000 lx, respectively. 

### 2.3. Data Analysis

Mortality rate (number dying in a stage divided by the number at the start), development data (time in stage in days) and generation time (the time from egg to when 1/2 adults are dead) were calculated using SPSS (version: 23.0). The effects of temperature treatments on the development time and survival rate were analyzed using the Generalized Linear Model (GLM). Survival rate and development time were the dependent variables. The daytime temperature, nighttime temperature and their interaction were the independent variables, respectively. In order to determine the optimal error distribution in the GLM, we calculated the AIC values of 8 models in SPSS and selected the lowest AIC value model for every trait in this study. Egg development time and larvae time were modeled with Ordinal logistic and pupal time, adult time and generation time were modeled with Ordinal probit. We modelled the larvae survival rate with the Tweedie log link and the pupae survival rate with linear functions. The Tukey–Kramer HSD test was used for significant differences between the treatments of the development time and survival rate.

A linear regression model (Equation (1)) was used to describe the relationship between the developmental rate and temperature; R^2^ indicated the degree of fit: (1)$$V\left(T\right)=a+bT\;$$
where a, b are the model parameters, *V*(*T*) is the developmental rate of selected stage (1/time in stage) and *T* is the temperature.

According to the effective accumulated temperature rule, the least-squares method is used to calculate the developmental starting temperature and effective accumulated temperature of the armyworm at each developmental stage. The calculation Formulas (2) and (3) is as follows:(2)$$C=\frac{\Sigma V\sum T-\Sigma V\Sigma T}{n\Sigma V^{2}-{\left(\Sigma V\right)}^{2}}$$
(3)$$K=\frac{n\Sigma VT-\Sigma V\Sigma T}{n\Sigma V^{2}-{\left(\Sigma V\right)}^{2}}$$

In the formula, *T* is the treatment temperature; *C* is the developmental threshold temperature; *K* is the effective accumulated temperature constant; *V* is the developmental rate (1/developmental days); n is the number of experimental temperature groups, in this study n = 9.

We used results from previous studies to compare the development among populations (see Table 1).

## 3. Results

### 3.1. Development Duration 

The development duration of eggs (Figure 2A), larvae (Figure 2B), pupae (Figure 2C) and generation time (Figure 2E) increased with the decreased daytime temperature and nighttime temperature. Moth longevity was reduced with a declining daytime temperature and the nighttime temperature increasing (Figure 2D). Daytime temperature treatments had extremely significant effects on the duration of all stages, except for adults (egg, *df* = 2, *χ*^2^ = 45.602, *p* < 0.001; larvae, *df* = 2, *χ*^2^ = 96.213, *p* < 0.001; pupae, *df* = 2, *χ*^2^ = 112.745, *p* < 0.001; adult, *χ*^2^ = 14.083; *df* = 2; *p* < 0.01; generation time, *df* = 2, *χ*^2^ = 169.127, *p* < 0.001; Table 1). Nighttime temperature levels had significant effects on the larvae (*df* = 2, *χ*^2^ = 28.879, *p* < 0.001), pupae (*df* = 2, *χ*^2^ = 16.002, *p* < 0.001) and generation time (*df* = 2, *χ*^2^ = 72.711, *p* < 0.001).There were significant interaction effects on the egg (*df* = 4, *χ*^2^ = 20.272, *p* < 0.001), larvae (*df* = 4, *χ*^2^ = 20.272, *p* < 0.001), adult (*df* = 4, *χ*^2^ = 9.826, *p* < 0.05) and generation time (*df* = 4, *χ*^2^ = 10.934, *p* < 0.05). Generation time was 41.4 d, 35.6 d and 35.5 d in the group with a daytime temperature of 27 °C and nighttime temperatures of 21 °C, 23 °C and 25 °C, respectively. Generation time was 57.6 d, 55.9 d and 50 d with a daytime temperature of 24 °C and nighttime temperatures of 18 °C, 20 °C and 22 °C, respectively. Meanwhile, generation time was 70.3 d, 65.5 d and 60 d at a daytime temperature of 20 °C and nighttime temperatures of 14 °C, 16 °C and 18 °C, respectively.

### 3.2. Survival Rate

All eggs survived and hatched and there were no moth deaths over the first three days of adult life. Mortality rate of larvae (Figure 2F) and pupae (Figure 2G) increased with the decreased daytime temperature and nighttime temperature range; much more so for pupae (Figure 2F). Significant effects were observed between the daytime temperature regime and mortality in larvae ((*df* = 2, *χ*^2^ = 12.626, *p* < 0.01)) and pupae (*df* = 2, *χ*^2^ = 96.651, *p* < 0.001; Table 2). Significant effects were observed between the nighttime temperature regime and mortality in pupae (*df* = 2, *χ*^2^ = 54.990, *p* < 0.001). Significant interaction effects were observed in pupae (*df* = 2, *χ*^2^ = 12.953, *p* < 0.001).

### 3.3. Development of Larvae, Pupae and Adults

Compared to previous studies, the estimated lower temperature thresholds for larvae and generation were lower in this study, and higher for the pupal stage. The estimated thermal requirement in degree–days of larvae was higher, but lower in the pupae and generation than the one reported in previous studies (Table 3).

Developmental rate, *V*(*T*) increased with increased daytime and nighttime temperatures. The slope of the linear regression model indicates the developmental rate of insects. The linear equation for larvae was *V*(*T*) = 0.0033T − 0.0423 (R^2^ = 0.8389), for pupae *V*(*T*) = 0.0113T − 0.154 (R^2^ = 0.9402) and generation *V*(*T*) = 0.0016T − 0.0123 (R^2^ = 0.6582), where *T* is the average temperature experienced. Compared to the results obtained from previously conducted research under constant temperature [26,29,30,31], the slope of the linear regression model for larvae and adults in this study was similar. The slope for pupae in our study was higher than in previous studies.

### 3.4. Pupation and Emergence Uniformity

Higher uniformity of *S. frugiperda* pupation (Figure 3A) and emergence (Figure 3B) were observed in high daytime temperature and nighttime temperature regimes. The start of pupation and emergence in the treatment of 27–25 °C were 20 d and 27 d, respectively, and 40 d and 60 d, respectively, compared to those in the treatment of 20–14 °C.

## 4. Discussion

In ectotherms, body temperature more or less tracks changes in ambient temperature, although behavior can greatly modify the temperature experienced [32,33,34,35]. The temperature experienced in the organism’s physical environment will strongly influence development, survival, potential and realized fecundity, and migration; indeed, virtually all aspects of its physiological ecology [36]. Invariably, when an insect species invades a new area, there is considerable interest in understanding these temperature effects, and *Spodoptera frugiperda* is no exception, e.g., in China [24,25,26,27,29,36], India [30,31,32,33,34,35,36,37,38] and Kenya [39]. These studies are locally focused and are generally based on constant temperatures. The premise and, indeed, promise is that, using these data, we can predict development based on a fitted model using laboratory results [40]. Surprisingly, constant temperature results, and even simple degree–day or linear models, and rate summing are informative, although they can be right for the wrong reasons. Much of the ability to predict results depends on what temperatures are experienced. In practice, real temperatures fluctuate, and various studies have reported such effects [41,42,43,44]. Here, we tested simple changes in the temperature experienced based on daytime–nighttime temperature changes relevant to the invaded range in tropical parts of southern China. In these regions, the shape of the temperature curve can indeed be more or less flat between small daytime–nighttime changes.

As expected, significant effects on development were observed in all temperature treatments; generally, the higher the temperature, the faster the development, but there were significant interaction effects as a result of temperature changes. The greatest effect of the fluctuation was the slowing down of development at intermediate temperatures (24 °C), relative to the simple expectations of a degree–day model. Fluctuations increased the rate of development at low temperature treatments (20 °C); so, in this treatment, the greater the fluctuation, the faster the development. Similar fluctuating temperature effects, namely increasing rates of development, have been observed in other studies [45]. There was little to no effect at near-optimal temperatures (27 °C). Linear degree–day models are meant to apply well to the intermediate temperature range in which our experimental treatments fell. However, predictions are likely to have an error margin of up to 5 days for larvae. The timing of management decisions may need to be adjusted for such effects of daytime–nighttime temperature changes.

Other studies have also shown that larval growth rate, survival rate, adult longevity and fecundity are moderately increased in nighttime temperatures [46]; and, for the California Red Scale, *Aonidiella aurantia* (Mask.), population density in the Covina Valley is inversely correlated with the number of nights with the temperature dropping [47]. Whitney and Johnson (2005) showed that increasing the nighttime temperature increased the intrinsic rate of growth in *Pieris rapae* [48]. The effects of nighttime temperatures on the adult lifespan of the Finnish Glanville fritillary butterfly (*Melitaea cinxia*) [49] were similar with this study. The larval and pupal survival was only significantly affected by the highest daytime temperature. Low nighttime temperatures had little effect on the survival and the overall maturation feeding of *Monochamus alternatus* adult beetles [50].

Geographical ranges, population dynamics and phenologies of many organisms are altered by climate change. For ectotherms, increased ambient temperatures frequently have direct consequences for metabolic rates, activity patterns and developmental rates. Consequently, in many insect species, earlier and prolonged seasonal duration have occurred with recent global warming. However, from an ecological and evolutionary perspective, the voltinism and investment in each generation may be even more important than seasonality, since an additional generation per unit time may accelerate population growth and the potential for adaptation [51,52,53]. The pupation and emergence of *S. frugiperda* increased along with the increased daytime temperature and nighttime temperature. The uniformity of pupation and emergence is higher in the warmer night in this study, indicating that the growth, development, population and voltinism will increase. We suggest that management of *S. frugiperda* will become more costly to the economy and the environment.

The Chinese government has announced the rules on *S. frugiperda* management for Hainan, Guangdong, Guangxi and Yunnan provinces, which are the winter reproducing areas of *S. frugiperda*. Monitoring the invaded population and decreasing the number of pest sources that may migrate to northern China from this area are two critical objectives [1,13,54]. However, as mentioned earlier, with the huge demand of fresh maize consumption in China, the winter maize planting area has increased rapidly in Guangxi, Guangdong and Hainan provinces. The population of *S. frugiperda* in this area may serve as a source population for the annual migration and invasion of northern China, but transgenic varieties and pesticides cannot be used, as sweet maize is directly consumed by people. Addressing *S. frugiperda* management will require a marked change in policy. In conclusion, the results of this study could help the local farmers to time biological and physical control interventions better.

## Figures and Tables

**Figure 1 insects-15-00180-f001:**
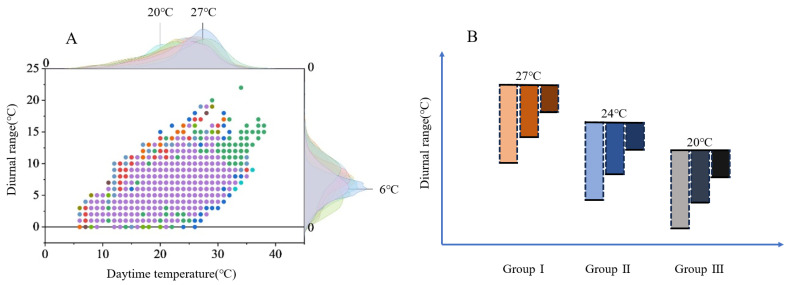
Daytime temperature and diurnal temperature range in Guangdong, Guangxi and Hainan from October to March in 2018–2022. Dots of different colors indicate the diurnal ranges of different daytime temperature (**A**); Design of temperature treatments used in this study (**B**), the nighttime temperature decreased from the diurnal temperature by 6 °C to 4 °C and 2 °C from left to right in each treatment group.

**Figure 2 insects-15-00180-f002:**
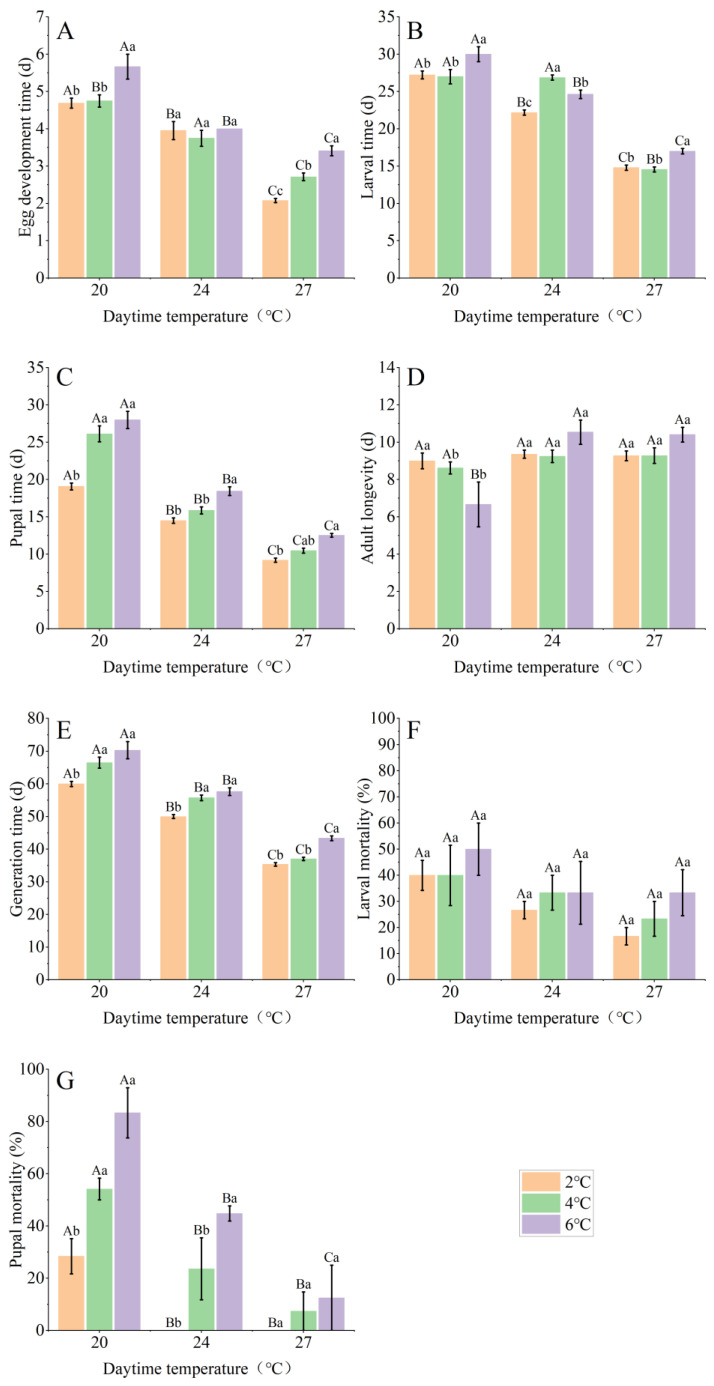
Development time (±SE) of (**A**) egg, (**B**) larvae, (**C**) pupae, (**D**) adult, (**E**) generation time, larval mortality (**F**) and pupal mortality (**G**) of *Spodoptera frugiperda* at different daytime–nighttime temperature regimes. Capital letters indicate the significance of the daytime temperature (*p* < 0.05), while lowercase letters indicate the significance between diurnal ranges (*p* < 0.05).

**Figure 3 insects-15-00180-f003:**
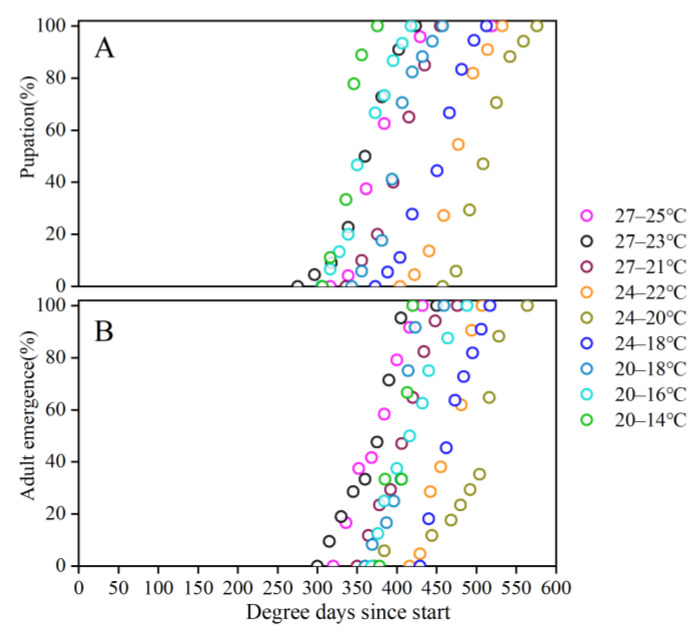
Cumulative percentage of pupation (**A**) and adult emergence (**B**) for each temperature regime plotted against degree–days experienced.

**Table 1 insects-15-00180-t001:** Effect of diurnal temperature regimes on the developmental time of *Spodoptera frugiperda*.

Trait	Source	*χ* ^2^	*df*	*p*
Egg	Daytime temperature	45.602	2	***p* < 0.001**
	Nighttime temperature	2.304	2	0.316
	Daytime temperature × Nighttime temperature	20.272	4	***p* < 0.001**
Larvae	Daytime temperature	96.213	2	***p* < 0.001**
	Nighttime temperature	28.879	2	***p* < 0.001**
	Daytime temperature × Nighttime temperature	37.652	4	***p* < 0.001**
Pupae	Daytime temperature	112.745	2	***p* < 0.001**
	Nighttime temperature	16.002	2	***p* < 0.001**
	Daytime temperature × Nighttime temperature	1.011	4	0.908
Adults	Daytime temperature	14.083	2	***p* < 0.01**
	Nighttime temperature	0.353	2	0.838
	Daytime temperature × Nighttime temperature	9.826	4	***p* < 0.05**
Generation time	Daytime temperature	169.127	2	***p* < 0.001**
	Nighttime temperature	72.711	2	***p* < 0.001**
	Daytime temperature × Nighttime temperature	10.934	4	***p* < 0.05**

Note: Significant *p*-values are given in bold.

**Table 2 insects-15-00180-t002:** Effect of diurnal temperature regimes on the survival rate in different stages of *Spodoptera frugiperda*.

Stage	Source	*χ* ^2^	*df*	*p*
Larvae	Daytime temperature	12.626	2	***p* < 0.01**
	Nighttime temperature	4.865	2	0.088
	Daytime temperature × Nighttime temperature	1.949	4	0.745
Pupae	Daytime temperature	96.651	2	***p* < 0.001**
	Nighttime temperature	54.990	2	***p* < 0.001**
	Daytime temperature × Nighttime temperature	12.953	4	***p* < 0.05**

Note: Significant *p*-values are given in bold.

**Table 3 insects-15-00180-t003:** Parameters for the linear regression model of development rate against temperature.

Studies	Larvae				Pupae				Generation Time			
	*C* ± S.E.	*K* ± S.E.	Slope	R^2^	*C* ± S.E.	*K* ± S.E.	Slope	R^2^	*C* ± S.E.	*K* ± S.E.	Slope	R^2^
This study	9.55 ± 0.66	296.88 ± 2.79	0.0033	0.86	13.59 ± 0.68	129.13 ± 2.26	0.0113	0.943	8.25 ± 0.99	551.89 ± 5.00	0.0016	0.89
Zhang et al., 2020 [26]	11.11 ± 3.15	201.25 ± 6.41	0.005	0.972	11.01 ± 2.17	134.12 ± 7.85	0.0075	0.812	9.21 ± 1. 46	636.53 ± 7.39	0.0016	0.986
He et al., 2019 [29]	11.10 ± 0.70	211.93 ± 2.55	0.005	0.948	11.92 ± 0.85	135.69 ± 3.30	0.008	0.891	9.16 ± 0.64	680.02 ± 2.06	0.002	0.829
Du Plessis et al., 2020 [30]	12.12 ± 0.24	202.66 ± 4.45	0.0049	0.95	13.06 ± 0.19	150.29 ± 2.79	0.0067	0.97	/	/	/	/
Prasad et al., 2020 ^1^ [31]	9.74	217.39	0.0046	0.975	10.22	172.41	0.0058	0.974	/	/	/	/

Note: *C* = estimated lower temperature threshold; *K* = estimated thermal requirement in degree-days. ^1^ Data of Prasad et al. (2020) in the above table was recalculated by fitting a linear regression model.

## Data Availability

Data used in this study are available from the corresponding authors upon reasonable request.

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
