# Peer review of "Night Warming Has Mixed Effects on the Development of the Fall Armyworm, Spodoptera frugiperda (Lepidoptera, Noctuidae), in Southern China"

_insects, 2024, doi:10.3390/insects15030180_

Round 1

Reviewer 1 Report

Comments and Suggestions for Authors

Xu et al. present the results of a simple laboratory study on the effect of fluctuating temperatures on the development and survival of S. frugiperda larvae and pupae. The study is relevant to the calculation of degree-day models used to predict pest occurrence in the field.

The manuscript has numerous serious issues and I cannot recommend that it be accepted for publication.

Major issues

         I.            The authors measured development time, but apparently omitted to determine the effects of fluctuating temperature on body weight (larvae or pupae) or body size (adults). This would have been valuable given that temperature affects feeding, assimilation and growth. Why was this valuable and easy-to-collect information omitted?

       II.            The authors state that they measured adult longevity (line 99). But I could not find any results on adult longevity in the Results section.

     III.            The sample sizes for larvae, pupae and adults should be clearly stated given that  we are not told of the prevalence of egg hatch in the experimental insects and important levels of mortality occurred in the larval stage.

    IV.            The variables analyzed are not clearly defined (survival, development).

      V.            There seem to be numerous errors in reporting the statistical values, indeed, most of the Chi2 and P values do not seem to match!

I have written numerous suggestions and numbered points on a scanned copy of the manuscript.

Numbered points (see scanned copy):

1a. The term corn is generally used by the USA, whereas the rest of the world tends to use the term maize (as you did on line 49). Whichever one you chose, I would suggest using just one common name for this plant throughout the manuscript.

1b. The term "daily temperature" can be interpreted to signify "Mean daily temperature" which is the average of the highest and lowest temperatures during a 24-hour period (see https://forecast.weather.gov/glossary.php?word=mean%20daily%20temperature). I would strongly recommend that the authors use the term "daytime temperature" to avoid any confusion. This would apply throughout the manuscript.

2. You need to explain the meaning of "intermediate treatments" in the Simple Summary.

3. More rapid development than what?

4. L36. All stages except the egg. Generation time is not a stage.

5. I would like to see details of the artificial diet used. Please could you include a list of the ingredients and their quantities used in a supplemental file?

6. Usually pupae or eggs are subjected to disinfection steps such as use of hypochlorite or formaldehyde solution? Was this the case here?

7. Larvae were moved twice a day onto fresh leaves? This is a lot of handling.

8. Was soil sterilized for use as a pupation substrate?

9. Supplemental material is usually labeled as Supplemental file S1 or Figure S1.

10. How did you define survival rate? What were you actually measuring here?

11. How did you define the development time variable? Was this the number of hours/days spent in each instar or stage or some other variable?

12a. How did you check the suitability of fit of your GLM models? Did you examine residuals or AIC values or some other indicator?

12b. You mean the Tukey HSD test was used to compare treatment means?

13. This P value is incorrect. Should be 0.007 according to my calculations.

14.  This P value is incorrect. Should be  0.039 according to my calculations.

15. P values of 0.000 are meaningless. The conventional way to express small P values is P < 0.001.

16. The experimental design is a two-way ANOVA with daytime temp as a factor and nighttime temp or diurnal fluctuation as another factor. Did you analyze it in this way? This was unclear to me in the Results section – is this why the df values are always 2 for the main factors?

17. How did you define generation time? Not explained in the methods section.

18. The Chi2 and the P values do not match up. This looks like a non-significant effect (Chi2 = 9.1, P = 0.057).

19. Again the P value appears to be incorrect and appears non-significant to me P = 0.064.

20. Again the P value should be non-significant for a Chi2 of 3.4, P = 0.48

21. Why have you plotted the treatment on the x-axis as DESCENDING values 27, 24, 20 C? ....x-axis values are conventionally plotted as increasing values.

I was expecting to see the results of the Tukey HSD post-hoc tests mentioned in Line 126 in these graphs. Why were they omitted?

22. Again, these Chi2 and P values appear to be erroneous.

23. Fig 3. The y-axis labels should not include the word "proportion" as they do not show proportions. See my suggested edits.

24. Right for the wrong reasons? This seems like a misleading statement. On lines 219-220 you state that linear degree day models are designed to apply well to normal temperatures. Your models are similar to those published previously using constant temperatures, so is there a real problem here? I accept that using fluctuating temperatures can reveal some "daytime x diurnal fluctuation" interactions that can improve accuracy, but you seem to be overly critical of these models.

25. "dropped" seems a vague term. Perhaps you could reword this sentence.

26. Suppl Figure 1. See suggested edits to axis labels on A and B.

Comments on the Quality of English Language

Minor editing required.

Author Response

Dear Reviewer,

    Thank you very much for your time and your invaluable revision comments. We have carefully studied all your comments and tried our best to revise the manuscript accordingly. We listed our responses to each of your comments in world document attached. Please refer to the line number in revised manuscript. For the purpose of easy tracking in the main text, we highlighted our revisions in green.

    Your comments have substantially helped us with improving the manuscript. Please kindly review the revised version. We look forwards to your feedback.

Sincerely,

Yangcheng Xu

Reviewer 2 Report

Comments and Suggestions for Authors

This paper investigates the effects of a variable temperature regime on the survival and development of the fall armyworm (Spodoptera frugiperda). This research enhances the current state of knowledge about how climate change may affect the success of this important crop pest. It begins to address a gap in much of the research that is published on this (and many other) insect species- the role of changing temperatures in determining developmental timing and biological success of the species. Many other studies use constant rearing temperatures, which is convenient for research design but ignores the real and impactful effects of changing temperatures on insects. Further, the authors of this study used variable temperature regimes that included a smaller range, mimicking some of the changes we are seeing, as anthropogenic climate change has had a disproportionate effect on low temperatures.

Overall, this is a straightforward study that brings up some important points about the current state of knowledge and reveals some significant impacts of a variable temperature regime on S. frugiperda development.

Lines 31-32- including the temperatures used in each treatment seems too granular to fit well in the abstract. Instead, I recommend emphasizing the diminishing range of temperatures used in these treatments, as that is the focus of the experimental manipulations. Including the actual temperatures used in this brief summary detracted from the underlying ideas being conveyed.

Lines 101-104- I think this information fits better in an introduction than in the methods. Indeed, it could be expanded into a standalone paragraph about the thermal tolerance of the species and how most of what is known doesn’t account for daily or weather-related variability in temperature.

Lines 112-114- Please provide justification for the design of your thermal regime. This seems like the weakest part of this study to me. Elsewhere, there is emphasis on how standard insect rearing conditions lack ecological relevance because of stable rearing temperatures. Yet your means to address this gap includes a rather artificial thermal regime, with stable temperatures except for a 30 minute period of warming or cooling between the diurnal and nocturnal cycles. While this is, no doubt, more accurate to the conditions experienced in nature, it is still a very artificial set of conditions.

Lines 201-210- “Reinvent a wheel” seems like an unnecessarily negative phrase to use here. Indeed, the whole paragraph is diminishing the value of previous work in this area too much. Previous studies on this (and other) pest insect that have used thermally stable rearing conditions have still provided really important information that is critical to management. Indeed, with a species that covers a large geographic area, is expanding its range, and has distinct genetic strains, understanding the baseline physiology of different populations within the same species is important. It does not seem appropriate to diminish the value of previous studies by using a phrase like this.

Lines 226- Please provide the genus and species for red scale.

Lines 224-227- This point is awkwardly worded. Please revise for clarity.

Figure 3- This figure appears to be presented in low resolution in my version of this manuscript. As this figure relies on differentiating among several small, differently colored circles, this should be included at higher resolution/quality.

Discussion- I would like to see a somewhat expanded discussion. The important advancement of knowledge that this paper presents is about looking at variable temperature regimes (mimicking the diurnal cycle of heating and cooling). While there is little published about development and success of S. frugiperda with a variable temperature regime, there have been several studies published on this area more generally. The authors mention some of these studies in the discussion, but only nominally. Additionally, since the goal of this paper is to gain a more accurate understanding of how ecologically relevant conditions affect S. frugiperda, it is worth noting that several abiotic cues in addition to temperature can have significant influences on insect growth and development. I do not think it is necessary to collect additional data investigating these parameters with variations in humidity, light cycles, etc., but it would be valuable to acknowledge these as factors that might be important to the story that is being told here. This would be particularly valuable if the authors could comment on the changes in light cycles, average humidities, UV exposure, etc. that are commonly experiences in these regions during the most relevant parts of the year. Additionally, this information will help contextualize the authors’ decisions around rearing and treatment humidity and light cycles used in this study.

Supplementary figure 1A- I am struggling to understand this figure as it is presented. Please clarify either with more information in the figure legend or by separating this into multiple panels in the figure.

Comments on the Quality of English Language

I would suggest some widespread and general editing. Throughout the manuscript, there are several grammar, punctuation, and syntax issues that made it difficult to read and understand. While not poorly written overall, some focused editing could dramatically improve the readability and accessibility of this study.

Author Response

(The authors gave the same response as above.)

Round 2

Reviewer 1 Report

Comments and Suggestions for Authors

The authors have markedly improved their manuscript although it still requires additional changes.

1. I was unable to find the Supplemental Figure S1 that the authors say that they have modified.

2. The reporting of F statistics is incomplete. F statistics are a ratio of treatment and residual (error) variation. As such, F values can only be understood if treatment AND error degrees of freedom are reported. The authors have reported treatment degrees of freedom but not the corresponding residual(error) df values.

This applies to the text in Section 3.1, Table 1, section 3.2 text, and Table 2.

3. I previously asked how the authors assessed the suitability of their GLM models. They state that this has been clarified but I could find no information. Please indicate how the fit of models was assessed.

Additionally, I suspect that some of the fitted models may have been quasibinomial or quasipoisson as there is high variation in some of the data shown in Fig 2. Please clarify.

Comments on the Quality of English Language

Minor editing needed.

Author Response

Dear Reviewer, 

    We feel great thanks for your professional review work on our article. We have recalculated the development time and survival rate data in section 3.1 and 3.2; table 1 and table 2; replotted the figure 2 according your comments. We tried our best to improve the manuscript and made some changes marked in green in revised paper which will not influence the content and framework of the paper. We appreciate for your warm work earnestly, and hope the correction will meet with approval. Once again, thank you very much for your comments and suggestions.

Sincerely,

Yangcheng Xu
